# Left Ventricular Papillary Muscle: Anatomy, Pathophysiology, and Multimodal Evaluation

**DOI:** 10.3390/diagnostics14121270

**Published:** 2024-06-16

**Authors:** Shiying Li, Zhen Wang, Wenpei Fu, Fangya Li, Hui Gu, Nan Cui, Yixia Lin, Mingxing Xie, Yali Yang

**Affiliations:** 1Department of Ultrasound Medicine, Union Hospital, Tongji Medical College, Huazhong University of Science and Technology, Wuhan 430022, China; lsy1999@hust.edu.cn (S.L.); whuwangzhen1993@163.com (Z.W.); fuwenpei@hust.edu.cn (W.F.); lify0727@hust.edu.cn (F.L.); celiagh@126.com (H.G.); m202376270@hust.edu.cn (N.C.); linyixia@hust.edu.cn (Y.L.); xiemx@hust.edu.cn (M.X.); 2Hubei Province Clinical Research Center for Medical Imaging, Wuhan 430022, China; 3Hubei Province Key Laboratory of Molecular Imaging, Wuhan 430022, China

**Keywords:** papillary muscle, mitral regurgitation, multimodal imaging, functional evaluation

## Abstract

As an integral part of the mitral valve apparatus, the left ventricle papillary muscle (PM) controls mitral valve closure during systole and participates in the ejection process during left ventricular systole. Mitral regurgitation (MR) is the most immediate and predominant result when the PM is structurally or functionally abnormal. However, dysfunction of the PM is easily underestimated or overlooked in clinical interventions for MR-related diseases. Therefore, adequate recognition of PM dysfunction and PM-derived MR is critical. In this review, we systematically describe the normal anatomical variations in the PM and the pathophysiology of PM dysfunction-related diseases and summarize the commonly used parameters and the advantages and disadvantages of various noninvasive imaging modalities for the structural and functional assessment of the PM.

## 1. Introduction

As an integral part of the mitral valve apparatus, the left ventricle papillary muscle (PM) controls mitral valve closure during systole and participates in the ejection process during left ventricular systole, which plays an important role in maintaining normal cardiac function. Nonetheless, most previous studies have concentrated on the clinical assessment and management of mitral regurgitation caused by anatomical or functional anomalies of the PM, frequently ignoring interventions aimed at the PM. Indeed, there is a broad clinical spectrum of PM [1]. From asymptomatic mitral regurgitation in ischemic PM dysfunction to severe acute mitral regurgitation with cardiogenic shock and acute pulmonary edema in PM rupture [2], from no effect on cardiac function to a contribution to the development of left ventricular outflow tract obstruction (LVOT) by variants such as PM hypertrophy and displacement [3], the importance of the PM is increasingly being emphasized. Therefore, evaluating the structure and function of the PM is essential.

It is typically thought that only two PMs, anterolateral and posteromedial, are in the left ventricle; however, autopsy findings indicate wide variation in the PM, from morphology to number, and such a typical form of two PMs has been reported to be present in only 3–25% of the population [4]. PMs are more movable during the cardiac cycle, which further increases the difficulty of evaluating them in the clinical setting using various types of imaging modalities. Currently, the multimodal image evaluation of the PM is receiving increasing attention from researchers. Cardiovascular magnetic resonance (CMR), with its excellent spatial and temporal resolution, inherent soft tissue contrast, and absence of ionizing radiation, has become the examination modality used in most studies to evaluate the PM [5,6,7]. However, echocardiography, the primary modality for assessing the morphological structure of the heart, has been reported relatively infrequently for the assessment of the PM. Based on our clinical findings, we believe that ultrasound has more strengths than CMR in assessing the morphology and number of PMs due to its advantages of multiangle, multisection, and dynamic exploration. In addition, cardiac-gated CT is also used for the assessment of PM structure, and SPECT is used to assess PM perfusion.

In this review, we will present the anatomical features of the PM in detail, discuss the pathophysiological changes associated with PM dysfunction, and summarize the commonly used parameters for the multimodal imaging assessment of the PM.

## 2. Normal Anatomy of the PM

### 2.1. PM Structure and Variation

As an integral part of the mitral valve apparatus, the PM is a muscular structure that originates from the ventricular wall. We agree that the PM directly connects to the solid portion of the left ventricle through an extensive base. However, in 2004, Prof. XLeon Axel [8] imaged the PM of 25 individuals by X-ray multidetector array CT and found that the PM was attached to the left ventricular wall through a network of interwoven trabecular muscle rather than originating directly from the left ventricular myocardium. They considered that this special form would affect the way forces are transmitted between the PM and the left ventricular wall.

In general, we believe that there are only “2” PMs in the left ventricle, which are the anterolateral and posteromedial columns (Figure 1A, Appendix A). However, the autopsy results show that the typical form described above is present in only 3.5–30% of the population, and in the vast majority of people, the anterolateral and posteromedial PMs are composed of more than one muscle column (Figure 1B–D, Appendix A) [4]; therefore, the quantity of the PM will be better described by using the term “group” rather than “each”. Interestingly, the autopsy results also revealed the presence of more than two groups of PMs—the accessory PM (Figure 1E, Appendix A)—which is present in approximately 30% of the population [9]. Sato, T. [10] et al. reported that the chordae tendineae emanating from the intermediate accessory PM innervate the empty area in the middle of the mitral leaflet where no chordae tendineae are attached, which provides a safe pathway for the introduction and deployment of a MitraClip; therefore, the presence of the intermediate accessory PM would make MitraClip surgery difficult.

In addition to the variable numbers, there is also a large variability in the morphology of the PM. The autopsy results revealed that conical, truncated, pyramidal, and fan-shaped structures were the most common shapes of the PM. In addition, there are also some special but common anatomical patterns, such as the single base and divided apex, bifurcated PM, and “separate bases and fused apex” of the PM (Figure 2A–D, Appendix A) [4]. To further complicate matters, the above variants can be combined in the same ventricle via random permutations.

In conclusion, the PM is highly variable in terms of number and shape; these variants can occur simultaneously within the ventricles of the same individual, and each individual possesses a unique PM spectrum.

### 2.2. Blood Supply of the PM

The left and right coronary arteries emanate from the root of the aorta and course along the surface of the heart from the base to the apical region of the heart, branching out into the myocardium along the way. The finer class A branches of these vessels are intertwined within the ventricular myocardium and supply the middle and outer one-third of the ventricular wall, whereas less branched class B vessels, which have relatively thicker diameters, enter the PM and emit secondary vessels along the long axis of the PM to supply different segments of the PM (Figure 3) [11].

According to the autopsy findings, the anterolateral PM was dually supplied by the left anterior descending artery (LAD) and left circumflex artery (LCX), whereas the blood supply of the posteromedial PM depended on the dominant type of coronary artery. The posteromedial PM is supplied by the right coronary artery (RCA) in the right coronary dominant heart and the LCX in the left coronary dominant heart. In general, the posteromedial PM is supplied by a single vessel [2,11]. Therefore, it is generally accepted that the posteromedial PM is more sensitive to ischemia than the anterolateral PM, according to a meta-analysis by Giulio Massimi et al. [12]. The incidence of ischemic posteromedial papillary muscle rupture (PMR) is approximately 2.3 times greater than that of anterolateral PM rupture. However, the findings of David Wendell’s team [13] in 2022 showed that in patients with myocardial infarction with single-segment coronary occlusion, the LAD was 1.7 times more likely to develop obstruction than the RCA, but infarction involved the posteromedial PM more frequently than the anterolateral PM. In response to this paradox, they further correlated the obstructed vessels with the infarcted PM, concluding that “it is because there are more coronary segments associated with posteromedial PM infarction than anterolateral PM”.

In summary, although extensive research on the PM vascular system dates back to 1969 [1,14], the specific reasons for the difference in sensitivity to ischemia between the two groups of PMs in the left ventricle need to be further investigated in large samples, and no studies have examined the blood supply system for the accessory PM.

## 3. Physiologic Functions of the Papillary Muscle

As an integral component of the mitral valve apparatus, the PM plays a crucial role in modulating the opening and closure of the mitral valve throughout the cardiac cycle. During systole, these muscles contract synchronously with the left ventricular myocardium, transmitting longitudinal tension generated by their contraction to the mitral valve through the chordae tendineae. The tension applied downward by the PM via the chordae tendineae on the mitral leaflets, preventing their prolapse into the left atrium during systole, is referred to as “tethering” [15]. This tethering mechanism maintains a balance among the forces facilitating the closure of both the anterior and posterior mitral leaflets, arising from the inward displacement of the PM due to the contraction of the left ventricular wall and mitral annulus. Consequently, this mechanism ensures that the mitral valve closes adequately without prolapse [16,17,18].

Moreover, the morphology, longitudinal axis orientation, and dynamics of the contraction and diastole of the PM are critical factors in preserving normal left ventricular ejection function [19,20]. The longitudinal axis of these muscles is generally parallel to the left ventricular long axis, and their orientation is nearly perpendicular to the mitral valve annulus [5]. This spatial relationship optimizes the transmission of forces generated during left ventricular wall and papillary muscle contraction to the chordae tendineae, facilitating efficient force transfer to the mitral valve leaflets. Additionally, the relatively small volume and characteristic growth along the ventricular long axis of the PM, constituting only 8.9 ± 1.4% of the left ventricular mass, contribute to maintaining sufficient blood flow during the diastolic phase [5,19]. This ensures adequate left ventricular filling.

## 4. Pathophysiology of Papillary Muscle Dysfunction

### 4.1. Ischemic Papillary Muscle Dysfunction

PM, the final myocardial segment perfused by coronary blood supply, shows heightened sensitivity to ischemia, with ischemia being the predominant etiological factor contributing to its dysfunction. When ischemic insufficiency affects the PM, patients often manifest a diverse range of clinical phenotypes, ranging from mild MR to potentially life-threatening acute massive MR resulting from PMR [21,22].

Ingo Eitel et al. [23] demonstrated a heightened probability of moderate to severe MR in myocardial infarction patients who experienced papillary muscle infarction (PapMI). Furthermore, the PapMI has been unequivocally established as a critical independent predictor for major adverse cardiac events after myocardial infarction. Notably, isolated PapMI seldom leads to MR, which occurs only when PapMI coexists with the enlargement of the ventricle and a lack of coordination in ventricular wall motion [24]. Additionally, a study exploring the impact of the subpapillary ventricular wall and PM ischemia on the occurrence of advanced (moderate and severe) functional mitral regurgitation (FMR) in patients with ischemic cardiomyopathy revealed that posteromedial papillary muscle ischemia was associated with advanced FMR (OR: 1.60). However, in multivariable analysis, when controlling for subpapillary left ventricular ischemia with dysfunction, the association between the FMR and posteromedial papillary muscle ischemia weakened (*p* = 0.074) [25]. Moreover, using a sheep model, Messas et al. [26] discovered that the acute ischemia of the left ventricular inferior wall can produce FMR. Interestingly, this FMR was significantly reduced when there was the concurrent ischemia of the posteromedial papillary muscle [16]. This could be attributed to a reduction in the chordal tethering force on the posterior leaflet of the mitral valve due to the inadequate contraction of the PM [26].

When ischemic infarction results in PMR, its affirmative role in the occurrence of MR is established. Once PMR occurs, it leads to the simultaneous prolapse of half of the anterior and posterior leaflets of the mitral valve, causing acute eccentric severe regurgitation. At this point, due to the inability of the left atrium to compensate for the sudden increase in blood volume, acute pulmonary edema may occur, ultimately leading to death [22,27,28,29].

The above findings provide ample evidence for the role of papillary muscle dysfunction in the occurrence and progression of MR. These findings also emphasize the importance of early recognition and appropriate intervention for the abnormal morphology and function of the subvalvular apparatus in MR.

### 4.2. Nonischemic Papillary Muscle Dysfunction

The foremost factors contributing to the nonischemic dysfunction of the PM are alterations in the spatial position of the PM and asynchronous contraction of the anterolateral and posteromedial PM, the former resulting from nonischemic left ventricular dilatation and structural abnormalities within the PM itself.

Dilated cardiomyopathy (DCM), the primary cause of nonischemic left ventricular dilatation, is associated with diverse levels of impaired papillary muscle contractile function compared to that of healthy controls. The degree of papillary muscle contractile dysfunction is notably correlated with the severity of MR [30,31]. In addition, annular enlargement and mitral leaflet tethering caused by the displacement of the PM due to LV dilatation are the main mechanisms of FMR [32]. This finding suggested that the FMR due to DCM is affected by both the spatial position and contractile function of the PM. The potential cause of impaired papillary muscle contractile function in DCM patients is likely attributed to the fibrotic remodeling of the PM [30].

Hypertrophic cardiomyopathy (HCM) is currently one of the diseases most closely associated with abnormalities in papillary muscle structure [33]. Hypertrophy, anterior displacement, and bifurcation of the PM are the most common morphological manifestations [3,34]. Some studies indicate that these specific PM morphologies exacerbate the “SAM” phenomenon, even occupying the LVOT directly during systole, thereby causing LVOT obstruction [34,35,36]. However, there is currently no research reporting whether papillary muscle contractile function differs between HCM patients and healthy individuals.

The asynchronous contraction of the anterolateral and posteromedial PM is another mechanism contributing to the occurrence of MR, often resulting from a lack of coordination in ventricular wall motion at papillary muscle attachment sites [37,38,39]. Ypenburget al. [39] monitored the contraction times of the PM in 25 patients who underwent cardiac resynchronization therapy (CRT) and exhibited immediate relief in MR using speckle tracking echocardiography. They observed a reduction in asynchrony between the anterior-lateral and posterior-medial PM from 169 ± 69 ms to 25 ± 26 ms in all patients (*p* < 0.001). Philipp E. Bartko et al. [37] analyzed myocardial motion spectra in 269 patients with chronic heart failure and functional mitral regurgitation (FMR). They observed a correlation between the degree of asynchrony among the PM and the severity of FMR. Additionally, after cardiac resynchronization therapy (CRT), the degree of synchrony among papillary muscles was associated with the extent of relief in FMR patients.

## 5. Multimodal Image Evaluation of PM

The papillary muscle (PM), a free-standing muscle structure, exhibits significant mobility throughout the cardiac cycle. This increased movement compared to the more stable ventricular wall makes it challenging to capture its complete morphology in a single imaging plane or slice. Consequently, this adds complexity to the quantitative assessment of its morphological parameters using various imaging techniques.

Echocardiography, as an imaging method that allows for multifaceted and dynamic real-time monitoring of all cardiac structures, is a better solution to this challenge than CMR, CT, etc., thus becoming the preferred imaging modality for assessing the morphology of the PM. It is essential that, in addition to observing the PM itself, two-dimensional echocardiography allows for a dynamic assessment of the entire mitral valve apparatus and ventricular wall motion closely associated with the PM. Therefore, Doppler ultrasound can be utilized to evaluate the hemodynamic characteristics of the heart, facilitating a comprehensive imaging assessment. Due to its high temporal and spatial resolution, as well as diverse functional imaging sequences, cardiac magnetic resonance imaging (CMR) has become the predominant imaging modality for studying the structure and function of the PM at present [5,13,25,40]. Computed tomography (CT) allows dynamic assessment of the PM by cardiac gating, but its inability to provide hemodynamic information and radiation exposure limits its use in clinical practice [41,42]. In addition, SPECT allows perfusion imaging of the PM to assess the extent of ischemia and infarction, providing value for clinical decision-making [43].

### 5.1. Assessment of PM Structure

The structural assessment of the PM focuses on its location, morphology, number of groups, volume, mass, and relationship to the mitral valve and LV wall.

On the echocardiogram, the normal PM is located in the middle one-third of the left ventricular wall, and the diameter of the short axis is 0.7 ± 0.2 cm [36]. Generally, when the anterolateral PM is displaced anteriorly or apically relative to its normal position, it is displaced closer to the septum or the LV apex, which is referred to as anterior PM displacement or apical displacement, respectively. Hypertrophy is considered when the PM cross-sectional diameter is greater than 1.1 cm [5,36,44]. Echocardiography is the best means of visualizing the morphology of the PM because of its ability to explore from multiple perspectives, including anatomical features such as morphologic variants of the PM itself (e.g., bifurcation, multiple heads, etc.), the muscle bundles connecting the PM and left ventricular wall, and the distance of the PM tip to the mitral annulus. For the number of groups of PMs, the short-axis view of the left ventricle is the optimal section to clearly demonstrate the distribution and groups of PMs on the left ventricular wall; the most common is two groups of PMs, followed by three groups of PMs, and four groups of PMs are rarer [10,36]. Transesophageal echocardiography (TEE) allows for intraoperative assessment of the PM and, in combination with 3D-TEE, provides a clearer visualization of the relationship between the PM and other cardiac structures, which can guide surgery [36,45]. On the other hand, PM can also be imaged by 3D ultrasound to obtain the volume and mass of the PM [46]. However, it should be noted that the complete visualization of the entire PM in the ultrasound view is a challenge for the sonographer and demands the quality of the patient’s acoustic window.

In contrast, due to its ability to make three-dimensional, high-resolution measurements and because it does not require assumptions about the heart’s geometry, CMR has become the standard for functional and anatomical assessment of the heart, including the PM. On this basis, the volume and mass of the PM have been obtained using short-axis curved cardiac magnetic resonance images obtained from steady-state free-forward sequences, and the PM mass has been found to have a significant effect on the LV volume and mass determined by cine magnetic resonance imaging [19,47]. In addition, Jan Bogaert et al. monitored the PM by CMR and reported that PM hypertrophy and apical displacement were associated with the remodeling of the left ventricle [6,48]. In addition, CMR is commonly used to assess the morphology and number of PMs [5,49]. PM mobility, as a dynamic morphological index, is often assessed quantitatively on high-resolution CMR images. In particular, in patients with HCM in which there is a bifurcation of the anterolateral PM but no significant hypertrophy of the interventricular septum, the hypermobility of the bifurcated papillary muscle is involved in the formation of the LVOT obstruction [49]. The measurement procedure was as follows: in the four-chamber view of the heart, a line was drawn along the epicardial surface of the lateral wall of the LV(line B), another line was drawn in the direction of the long axis of the anterolateral PM (line A), and the change in the angle between the two lines during the cardiac cycle was considered the PM mobility (Figure 4).

### 5.2. Assessment of PM Function

#### 5.2.1. Qualitative Assessment of PM Function

A qualitative assessment of papillary muscle function is primarily an assessment of ischemia, infarction, and fibrosis using noninvasive imaging modalities. CMR, with its multiple functional imaging sequences, is currently the best imaging modality for the qualitative assessment of papillary muscle function. The noninvasive examination of necrotic myocardium with high signal density by late gadolinium-enhanced magnetic resonance imaging (LGE-MRI) is now recognized as the gold standard for identifying myocardial necrosis and myocardial viability [50]. Tanimoto et al. [24] imaged the hearts of patients with ST-segment elevation myocardial infarction (STEM) with LGE-MRI to evaluate the frequency of PapMI in these patients. They found that among 118 STEMI patients, LGE-MRI suggested the presence of PapMI in 40% of the patients. However, its accuracy is unknown, and there is often poor contrast between hyperenhanced PapMI and bright blood pools in cardiac chambers, which may lead to missed diagnoses. Flow-independent dark-blood delayed enhancement (FIDDLE) cardiac MRI, based on LGE-MRI, can suppress the bright blood pool signal and retain only the high signal of necrotic myocardium, thus improving the detection rate of myocardial infarction [51,52]. In 2022, Wendell et al. [13] verified in a canine infarction model that the accuracy of the FIDDL for the detection of PapMI was fully consistent with the pathologic findings. Using FIDDLE as the gold standard, they reported that the prevalence of anterolateral and posteromedial PapMI in patients with infarction was approximately 37% and 44%, respectively. In addition to its application in treating ischemic papillary muscle disease, Kozor et al. [7] used T1 mapping sequences to detect sphingolipid deposition in the LVPM of FD patients, which showed low signal intensity, to explore the mechanism of asymmetric hypertrophy of the PM in FD patients.

PET can detect myocardial perfusion and flow reserve. In the present study, nitrogen-13 (N-13) ammonia positron emission tomography (NH_3_ PET) was used to perform myocardial imaging to assess PM ischemia in 300 patients with suspected CAD, and after 910 days of follow-up, PM ischemia was present in 11% of the 30 patients who developed adverse cardiovascular events [43].

#### 5.2.2. Quantitative Assessment of PM Function

Papillary muscle contractility during the cardiac cycle is the main focus in the quantitative assessment of papillary muscle function (Table 1). The evaluation of contractile function is divided into longitudinal and radial contraction.

On the echocardiogram, for the assessment of longitudinal contractile function, the degree of PM contraction is primarily reflected through longitudinal strain, while the peak time of peak systolic PM strain (PT) reflects the speed of contraction. The most commonly used method for assessing PM longitudinal strain is the two-dimensional strain. This type of strain is referred to as the Lagrange strain, which is calculated based on the distance between two points in the myocardium during the systolic and diastolic phases [53]. PM longitudinal strain (PMls) is computed as PMls = (end diastolic PM length − end systolic PM length)/end diastolic PM length × 100%). This parameter is simply obtained from 2D ultrasound and CMR images. Tissue Doppler imaging (TDI) is another method for assessing PMls. It involves measuring the maximal PM length on a 2D echocardiogram and using the tissue Doppler mode to place the sample volume at the midpoint of the PM’s main axis. The strain rate curves from three cardiac cycles are averaged and integrated over time to derive the strain curves, from which the peak systolic strain of the PM is measured [54]. With the widespread use of 2D speckle tracking echocardiography (2D-STE) for myocardial function assessment, research using this technique for PM function assessment has progressively developed. In addition, 2D-STE can better overcome the difficulty of accurately assessing large PM motility due to its angle-independent features (Figure 5) [55,56,57]. 

**Table 1 diagnostics-14-01270-t001:** Parameters commonly used for quantitative assessment of the LVPM.

Aim of Study	Population	Groups	Index	Imaging Modality	Papers
assess the diagnostic performance of nT1 and PMls in the identification of iPPM.	46 STEMI patients	iPPM (n = 16) niPPM (n = 30)	PMls, nT1	CMR	Pambianchi, G. et al., 2023 [40]
investigate whether IPMD can affect MR severity independently of PM tethering distance in patients with LV dysfunction	83 patients with LV dysfunction (LVEF < 50%)	with FMR (n = 37) without FMR (n = 46)	IPMD, PM tethering distance	volumetric multislice computed tomography	Kim, K. et al., 2014 [41]
test whether PM dysfunction attenuates ischemic MR in patients with LV remodeling	40 patients with a piMI but without other lesions	Only with significant bulging at basal inferoposterior LV Without significant LV bulging	PM tethering distance, PMls	2D echocardiography + Tissue Doppler strain imaging	Uemura, T. et al. 2005 [54]
evaluate PMls in HCM and to find whether it has a value for prediction of SCD risk score	55 HCM patients	55 HCM patients 45 health controls	PMls	2D speckle tracking echocardiography	Koyuncu, A. et al., 2023 [55]
evaluate and compare PM function in and between patients with severe DMR and FMR	64 patients with severe MR	DMR (n = 39) FMR (n = 25) Health controls (n = 30)	PMls, circumferential strain	2D speckle tracking echocardiography	Kilicgedik, A. et al., 2017 [56]
assess the PMls as a contributor to recurrent MR after mitral valve repair for fibroelastic deficiency	64 patients with isolated posterior MVP and severe MR referred for surgery between 2008 and 2012	Post repair no MR (n = 56) Post repair recurrent MR (n = 8)	PMls	2D speckle tracking echocardiography	Grapsa, J. et al., 2015 [57]
assess PM and mitral valve structure and function in children and young adults with mild and moderate HCM	20 HCM patients	20 HCM patients 20 health controls	Apical PM displacement index	3D chocardiography	Joseph, N. et al., 2021 [46]
	524 consecutive patients who survived CABG and restrictive annuloplasty between 2001 and 2010	No MR Group (n = 412) MR Group (n = 112)	DYS-PAP, peak times	2D speckle tracking echocardiography	Van Garsse, L. et al., 2013 [58]
find a relationship between the level of PM asynchrony and the degree of MR in patients with ICM and nICM	31 patients underwent CAA with clinically advanced HF	ischemic cardiomyopathy (n = 21) nonischemic cardiomyopathy (n = 10)	DYS-PAP	Tissue Doppler strain imaging	Kordybach, M. et al., 2011 [59]
investigate the impact of impaired lateral shortening in the IPMD on mitral valve geometry and function in ischemic heart disease	67 patients with ischemic heart disease confirmed by cardiac catheterization underwent CMR	mild MR (n = 26) moderate/severe MR (n = 41)	IPMD	CMR	Kalra, K. et al., 2014 [60]
evaluate the relationship between papillary muscle T1 time and MR in DCM patients	40 DCM patients and 20 healthy adults	DCM with MR (n = 22) DCM without MR (n = 18) Health control (n = 20)	nT1	CMR+	Kato, S. et al., 2016 [30]

MI: myocardial infarction; iPPM: papillary muscle infarction; niPPM: nonpapillary muscle infiltration; PMls: PM longitudinal systolic strain; CMR: cardiac magnetic resonance imaging; nT1: native T1-mapping; LV: left ventricular; LVEF: left ventricle ejection fraction; FMR: functional mitral regurgitation; IPMD: interpapillary muscle distance; piMI: previous inferior myocardial infarction; HCM: hypertrophic cardiomyopathy; SCD: sudden cardiac death; DMR: degenerative mitral regurgitation; MVP: mitral valve prolapse; CABG: coronary artery bypass grafting; CAA: coronary artery angiography. HF: heart failure; DYS-PAP: papillary muscle systolic dyssynchrony; DCM: dilated cardiomyopathy; LVPM: left ventricle papillary muscle; STEMI: ST-segment elevation myocardial infarction; ICM: ischemic cardiomyopathy; nICM: nonischemic cardiomyopathy.

TDI and 2D-STE were able to obtain the PMls along with PTs [39,55]. Differences in PTs between the anterolateral and posteromedial PM contribute to the quantitative diagnosis of papillary muscle contraction dyssynchrony (DYS-PM), which has been suggested to be an independent predictor of MR recurrence after mitral valvuloplasty in ischemic MR [58]. However, there is no consensus on the definition of DYS-PM. Some researchers advocate that a difference of 65 ms or more in PTs between PMs can be considered an indication of DYS-PM, which is consistent with the criteria for LV contraction dyssynchrony. It has also been suggested that DYS-PM can be defined as a statistically significant difference in PTs between PMs [58,59]. The precise definition of DYS-PM still requires more research and discussion to provide clearer guidance for clinical diagnosis.

There is no standardized assessment of the transverse contractile function of the PM. The main parameters reported in the current literature include the interpapillary muscle distance (IPMD) and peak circumferential strain (ε strain), as IPMD refers to the shortening fraction from diastole to systole. It is generally recognized that reduced IPMD in patients with FMR and thus increased valve leaflet tethering are major determinants of the degree of FMR [41,60]. Measurements of IPMD are generally obtained in LV short-axis views at three levels: the root, middle, and tip of the PM. CMR and CT, as tomographic imaging modalities with manually adjustable layer thicknesses, are more accurate and convenient for the assessment of IPMD than echocardiography. The ε strain is the deformation of the PM in the radial direction of the LV, generally obtained by 2D-STE [59]. However, the clinical significance of the ε strain needs to be further explored.

The quantitative assessment of PM function is mainly based on two-dimensional (echocardiography, CMR, and CT) static images and TDI measurements. Moreover, 2D-STE, with the advantages of overcoming angle dependence and autonomous tracking, is the best way to dynamically assess the PM, but due to the large variability of the PM morphology and the influence of the patient’s acoustic window, there is no standard for a 2D-STE assessment strategy and a normal reference value range for the PM.

In conclusion, the current assessment of the PM has focused mainly on assessing its morphological structure. There is a lack of standardized guidelines for assessment strategies and normal reference ranges for functional assessment.

## 6. Conclusions

The normal anatomy and function of the PM are essential for normal cardiac pumping and maintenance of mitral valve function. Abnormalities in its anatomy occur mainly in patients with HCM, resulting in LVOT obstruction and mitral regurgitation; functional abnormalities often occur in ischemic cardiomyopathy and are associated with functional mitral regurgitation. Therefore, an understanding of the pathophysiology and imaging assessment of PM abnormalities is essential for the management of PM-related diseases. CMR can accurately assess the morphology and function of the PM and is currently the main imaging modality for studying the PM. Echocardiography can dynamically observe the morphologic structure and activity of the PM and can reveal functional parameters such as the PM strain through STE, making it a first-line imaging modality for assessing the PM; however, it requires high image quality and experience from ultrasonographers, and there is a lack of standardized assessment strategies for the functional assessment of the PM. SPECT can be used to assess perfusion in the PM, but few studies on this topic have been reported. In the future, with the continuous development of multimodal imaging and radiomics, the structural and functional assessment of the PM will become more operational and convenient, which will lead to the establishment of standardized assessment strategies and guidelines for normal reference ranges for the PM.

## Figures and Tables

**Figure 1 diagnostics-14-01270-f001:**
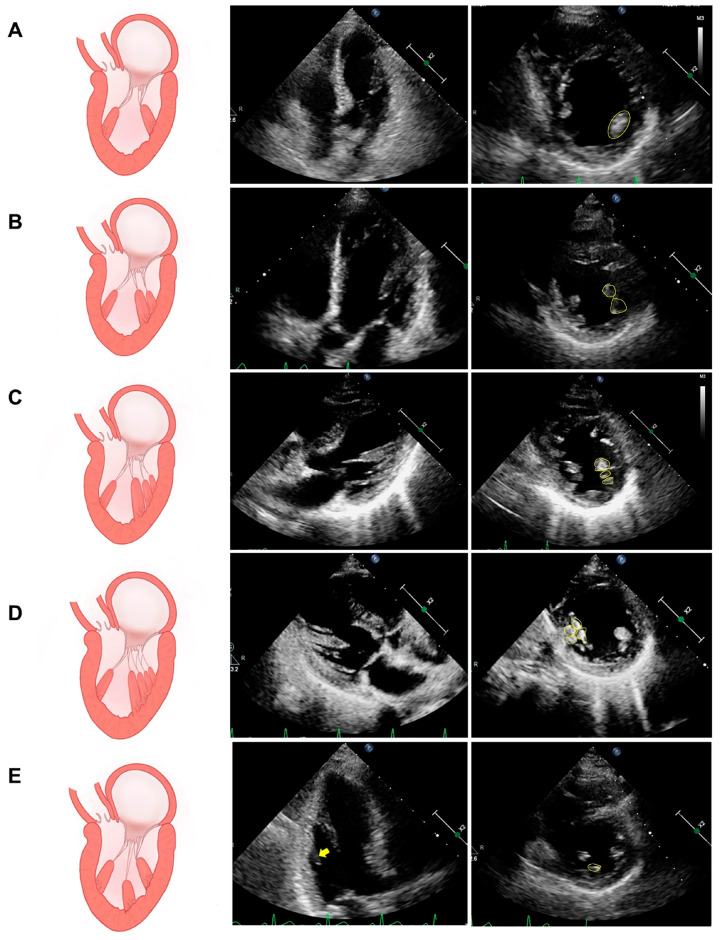
Schematics and echocardiograms of common variants in terms of the number of PM columns. (**A**) A typical PM with a single muscle. (**B**) Two parallel muscle columns of the PM. (**C**) Three parallel muscle columns of the PM. (**D**) Four parallel muscle columns of the PM. (**E**) The accessory PM (i.e., intermediate PM, the yellow arrow).

**Figure 2 diagnostics-14-01270-f002:**
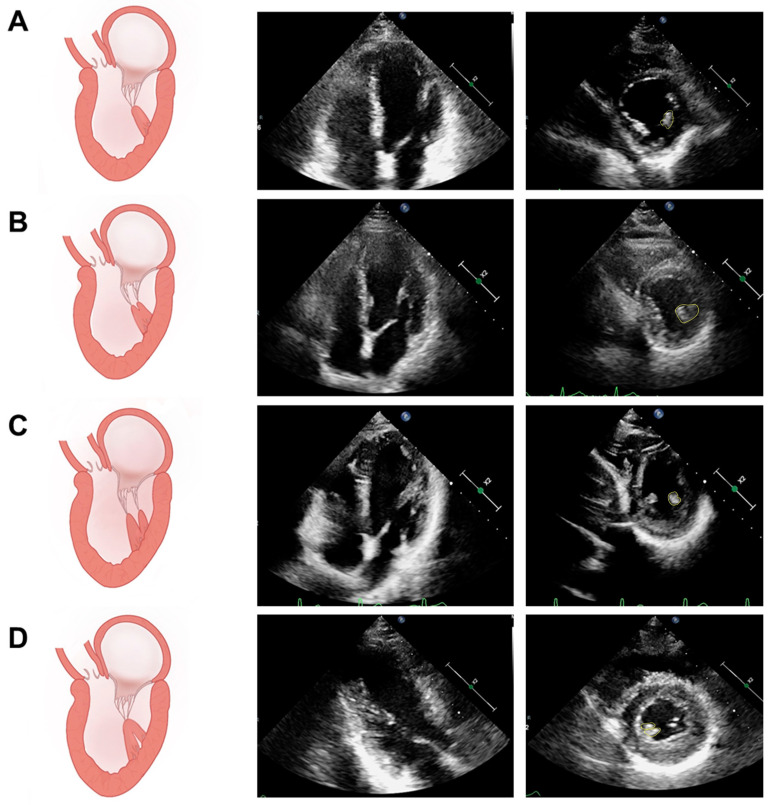
Schematics and echocardiograms of common variants in the shapes of PM columns (examples of posteromedial PM). (**A**) Typical column. (**B**) Single base and divided apex. (**C**) Bifurcated PM: single base with 2 or more muscle columns. (**D**) Separate bases and the fused apex. (The yellow circle indicates a muscle column).

**Figure 3 diagnostics-14-01270-f003:**
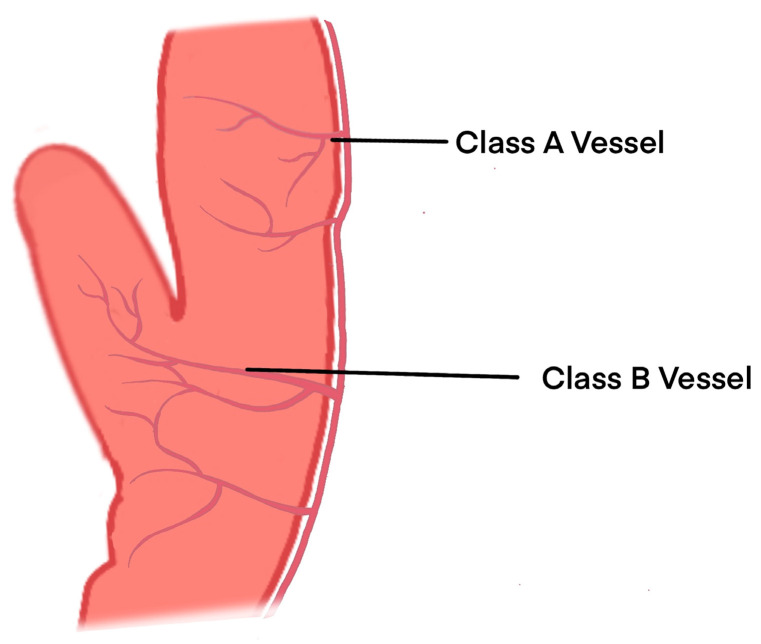
Schematic diagram of the microvascular distribution of the PM.

**Figure 4 diagnostics-14-01270-f004:**
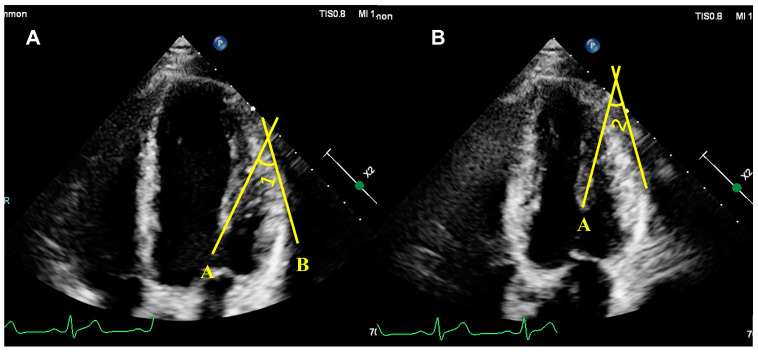
Measurement of papillary muscle mobility: changing angle between the long axis of the PM and the lateral wall of the left ventricle (**A**): at the end of diastole (∠1); (**B**): at the end of systole (∠2). PM mobility = ∠1 − ∠2.

**Figure 5 diagnostics-14-01270-f005:**
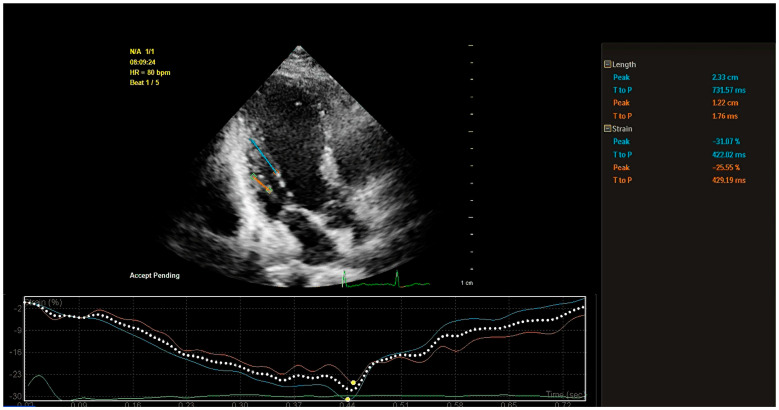
Assessment of longitudinal strain in the papillary muscle using 2D-STE on a Qlab workstation. The blue and orange lines represent the evaluated muscle segments. The right information panel displays the length and strain of the papillary muscle. The main focus is on the strain information, where “peak” indicates the peak systolic strain and “T to P” indicates the time to peak strain. In this figure, the peak systolic strains of the two muscle columns of the posteromedial papillary muscle were 31.07% and 21.55%, respectively, with time to peak strains of approximately 422.02 ms and 429.19 ms, respectively.

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
