# Peer review of "Left Ventricular Papillary Muscle: Anatomy, Pathophysiology, and Multimodal Evaluation"

_diagnostics, 2024, doi:10.3390/diagnostics14121270_

Round 1

Reviewer 1 Report

Comments and Suggestions for Authors

Page 5 paragraph 4: The paragraph on ischemic papillary muscle dysfunction is unclear.  Can the authors better explain what is meant by “advanced functional mitral regurgitation”.  The sentence:  “The observation may suggest that, in ischemic MR patients, the impaired contractile function attributed to papillary muscle ischemia somewhat mitigates the severity of MR” is difficult to comprehend.

Page 7 paragraph 1:  In the paragraph on papillary muscle dyssynchronization:  How does papillary muscle approximation improve outcome of mitral valve surgery, in the presence of papillary muscle dyscnchronization?

Page 7:

The sentence in the part 5: Pathophysiology of the papillary muscle dysfunction:  “The PM, as a part of the myocardium, is best characterized by greater mobility in the cardiac cycle than in the ventricular wall, which poses a challenge for the quantitative 220 assessment of its morphological parameters” is difficult to understand. .  How can you compare cardiac mobility in the cardiac cycle, ie a timed event, to the ventricular wall?

Page 9:  The paragraph on quantitative assessment of papillary muscle dysfunction is difficult to comprehend.  The is also perhaps too detailed for this manuscript.

Table 1 is too confusing with too many abbreviations, and is difficult to know how this contributes to the manuscript.

The entire paragraph for Figure 5 is confusing, too detailed, and  difficult to follow. The Figures itself is difficult to visualize. 

Comments on the Quality of English Language

Many sections of the manuscript were difficult to comprehend. I did note specific paragraphs to the authors that need to better written

Reviewer 2 Report

Comments and Suggestions for Authors

The authors of the article were very careful in searching and structuring the material. There are two minor comments, after correction of which the article can be considered for publication.

1. Reduce and update the list of references (more than half of the sources are older than 5 years ago)

2. Table 1 should be made according to the standard model, indicating the names of the researchers and the years of publication of the articles

Round 2

Reviewer 1 Report

Comments and Suggestions for Authors

This evaluation of papillary muscle anatomy, and function is very important, and this is a very important topic.